# Use of Health-Promoting Food and Supplements in Swiss Children

**DOI:** 10.3390/children9121842

**Published:** 2022-11-28

**Authors:** Corinne Légeret, Clarissa Lohmann, Laura Pedrini, Laurin Sarbach, Raoul Furlano, Henrik Köhler

**Affiliations:** 1University Children’s Hospital Basel, Spitalstrasse 33, 4056 Basel, Switzerland; 2Children’s Hospital of Aarau, Tellstrasse 25, 5001 Aarau, Switzerland; 3Medical Faculty, University of Basel, 4001 Basel, Switzerland

**Keywords:** children, supplements, vitamins, protein, lifestyle

## Abstract

Introduction: Our diet is the sum of many different influences and has visibly changed over the past decades. Since children also imitate their parents when it comes to eating habits, the aim of the study was to assess the current dietary habits in Swiss children. Method: Cross-sectional study of children between 0 and 18 years of age in Switzerland. Results: A total of 1964 children participated, with an average age of 7.4 years. A total of 57.9% of participants stated to buy supplements to promote health, while fruit juices/smoothies were the most popular product (42.5%), followed by protein-enriched products (40%) and vitamins/minerals (29%). A statistically significant correlation between longer screen time, a higher socioeconomic background, and the intake of supplements was found. Over 20% of all families regularly consume plant-based drinks. Discussion: This Swiss cross-sectional study of over 1900 participants reveals that 58% of all participants buy supplements or special kid’s food to promote the child’s health. There is a correlation between higher screen time, higher parental income, and the usage of supplements. A total of 23% of participating families consume at least one plant-based drink on a regular basis. As more and more families use supplements, the pediatrician should not only focus on weight, which reflects the intake of macronutrients but should also take a history of whether children omit certain foods or take supplements to ensure the child does not have a deficiency of micronutrients.

## 1. Introduction

The intake of food does not only serve pure life support but has various further aspects. Above all, there is the social part; it is culturally shaped, has a historical background, and, in addition, financial and geographical opportunities also influence our eating behavior, just to name a few [1]. That our diet is subject to multifactorial influence has never been so easy to observe than in veganism. While vegans have often faced ridicule and exclusion [2], it has gone mainstream in the last decade. In 2018 and 2019, the sales of plant-based foods grew by 29% in North America [3]. This quick change appears to have several causes, as it attracts people with an interest in slowing climate change and those with a health consciousness [4]. Most of all, it is mainly popular among a younger, financially strong, urban cohort, as many celebrities and influencers advertise such a diet [5]. Children imitate the eating behavior of their parents [6]; therefore, their diet seems to have changed over the last decade as well [7]. The aim of this study was to gain insight into the current eating behavior of children and adolescents and to relate it to socioeconomic background and leisure behavior.

## 2. Materials and Methods

This is a cross-sectional study between January and May 2022. Children between the age of 0 and 18 years were included. Questionnaires were handed out by instructed medical staff in the Children’s Hospital of Aarau and Basel and in four different pediatric practices in Switzerland. Participation was voluntary and anonymous. Exclusion criteria comprised a place of residence outside the Northwestern part of Switzerland, recently arrived refugees, hospitalized oncological patients, psychiatric patients, as well as critically ill outpatients in the emergency room.

### 2.1. Questionnaire

The questionnaire was created in German and translated into English and French and into the top ten languages. An interpreter was booked for the Children’s Hospital of Aarau in 2020 (Tigrinya, Albanian, Arabic, Farsi, Tamil, Dari, Turkish, Kurdish Sorani, Italian, and Portuguese). Only bilingual translators were used, and except for Tigrinya, Farsi, and Kurdish Sorani, an independent backward translation was performed to ensure accuracy. A first version of the questionnaire was distributed to 14 patients and tested for usefulness and clarity of the questions, which were slightly altered based on the answers. In the final questionnaire following questions were asked: age, sex, place of birth of the child and the parents, population of residence (city, locality with >10,000 inhabitants, rural locality or village), the level of education of both parents (less than 7 years of school, mandatory school, pre-apprenticeship, vocational apprenticeship or vocational school, high school, vocational baccalaureate or diploma school, higher technical and professional education, university or technical college, and not determinable, unknown), the amount of screen time (in hours/week), the use of a social media account and general physical activity and activity in a sport’s club (in hours/week). Information about underlying diseases and known allergies was collected. Participants were asked if supplements/foods are bought to promote the health of the child/adolescent. The following 7 categories were available: 1. effervescent tablets/lozenge bonbons containing vitamins, trace elements, or minerals; 2. sweets enriched with vitamins, trace elements, or minerals; 3. orange juice or smoothies; 4. products enriched with probiotic cultures; 5. special kid’s food (brands promoting healthy food for children (e.g., biscuits, crackers, bars, yogurts, drinks); 6. food compote in squeeze bag; and 7. protein-enriched products (powder, drinks, cottage cheese or packaging promotes ‘high in protein’). Additional information about the family consumption of milk and milk alternatives on a regular basis (defined as at least once a week) and the properties judged to be important when buying products (healthy, regional, organic, containing less sugar, without additives or individual answer) was collected. The questionnaire was designed to be very simple and clear to avoid (cultural) misunderstandings.

### 2.2. Definitions

The following answers were necessary for the questionnaire to count as filled out: age, sex, and a statement on the purchase of supplements/foods to promote health. Place of residence was defined as urban or rural (>10,000 or <10,000 inhabitants). The family background was classified as abroad once at least one parent was not born in Switzerland. Education of parents: For this variable, the higher-educated parent was chosen, and the category was divided into two groups (≤high school and >high school). Since there is no age-specific classification of low/normal/high screen time, we used 45 min/day as the cut-off for the toddlers and also performed calculations in 5 h steps per week for the older children to gain an overview of the children’s screen behavior.

### 2.3. Statistical Analysis

Univariate and multivariate analyses using χ^2^ test and logistic regression were applied with the exposure variables age, highest education of parents, screen time, and general activity in all age groups (0–3-year-olds, 4–9-year-olds, and 10–18-year-olds), activity in sport’s club in the two older age groups, the owning of a social media account in the oldest age group and the outcome variable purchase of supplements/foods to promote health. *p*-values < 0.05 were considered statistically significant. All data were analyzed with RStudio 2022.02.2.

### 2.4. Ethical Statement

The present study was conducted in accordance with the ethical principles laid down in the Declaration of Helsinki and its later amendments. Furthermore, it was approved by the local ethical committee (Ethics Committee of Northwest Switzerland, EKNZ, trial number 2021-02287).

## 3. Results

The questionnaire was distributed to 2471 children/adolescents or their parents, of which 1964 participated (response rate 79.5%). Six participants were excluded (four were older than 18 years, one lived outside of Switzerland, and one was a Ukrainian refugee); therefore, a total of 1958 questionnaires were included. A total of 153 (7.8%) included questionnaires were in another language than German (47 in English, 35 in French, 21 in Italian, 14 in Albanian, 11 in Turkish, 8 in Portuguese, 7 in Arabic, 6 in Tigrinya, 2 in Tamil, and 1, respectively, in Farsi and Dari). A total of 900 questionnaires were filled out at the Children’s Hospital of Aarau, 805 at the Children’s Hospital of Basel, and 253 in the pediatrician practices. Participants were, on average, 7.41 years old; for further characteristics, we refer to Table 1.

A total of 1133 (57.9%) participants stated buying supplements or food to promote the health of their child/themselves. Orange juice or smoothies was the most common category overall (42.5%), followed by products enriched with proteins (40%) and effervescent tablets/lozenge bonbons with vitamins, trace elements, or minerals (29.1%). The percentages of bought supplements in the age groups are shown in Figure 1.

The information about the child’s screen time and general activity in all age groups and activity in sports clubs and the use of a social media account in the older age groups are summarized in Table 2.

There was a statistically significant positive correlation between higher screen time of the children and the purchase of supplements/food to promote health in all age groups. The other values showed varying or no statistically significant effect, see Table 3.

A total of 96.8% of participants include cow’s milk in the family diet: 91.5% traditional cow’s milk, 4.85% only lactose-free cow’s milk, and 0.5% only raw cow’s milk. Only 2.71% of participants stated not consuming cow’s milk at all: 2.2% consume only plant-based drinks, 0.25% only goat, sheep, and/or buffalo milk, and 0.25% no milk or milk alternative at all. A total of 22.6% of participants’ families regularly consume at least one plant-based drink. The consumption of milk and milk alternatives is presented in Figure 2.

The most important properties of bought products (entered from suggestions and entered in blank fields) for the participants are shown in Figure 3: 4.7% stated that no particular property is important for them.

## 4. Discussion

This cross-sectional study investigated the intake of supplements in children/adolescents and revealed the following key findings: 58% of all participants stated buying supplements/food to promote health, special kid’s food, vitamins, and protein supplement being the most popular ones, screen time and higher income correlates with the purchase of supplements and 23% of participating families regularly consume (equivalent weekly) at least one plant-based drink.

In the group of children under 3 years of age, the most popular ‘food to promote health’ is special kid’s food, which was defined as fruit pouches, biscuits, crackers, etc., designed for children and is bought by 26% of all participating families. An Australian study showed the impressive global expansion of the baby food industry [8], which is explained by the fact that preparing fresh meals does take time, and family lives do become busier. Boak et al. [9] examined the parental motivations for making food choices for their infant children. Besides beliefs and values, mainly costs, availability, and parents’ capacities influence the choice. That health might be compromised for convenience when it comes to choosing food for toddlers was also shown in an American study [10]. Parents listed ‘child enjoys taste’ as the most common reason when choosing to purchase toddler-specific convenience foods, which 85% of participants consume regularly. Parents with a lower income reported that the presence of vitamins in toddlers’ convenience food is an important motivation to buy it. Since there are no Swiss data on this topic, we refer to a study from one of our neighboring countries: an Italian study interviewed over 360 parents regarding their choices of children’s food [11]; it was shown that especially women, parents with a higher income and higher level of education are more keen to buy functional foods for their children, a total of 36% of all parents claim to buy children’s food regularly, vitamin-enriched fruit juices are the most popular ones.

To attract children, convenience foods are sold in diverse forms and colors, in easy-to-use squeeze food pouches with plastic nozzles and taste very sweet, which is the secret of its worldwide success. Children are driven to the products by playful designs, but from a nutritional point of view, most products targeted at children are not adequate for them as they usually contain high energy density and an excessive content of added sugar and fat, as shown by studies from Portugal and England [12,13].

Concern has been raised as those fruity pouches contain much fructose (carbohydrate), which has not always been declared as ‘added sugar’ [14]. Several studies [15,16,17] were published showing that the median total sugar content of kids’ pouches was more than three times that of cans and jars. Concerns are not only a high energy density, but it may also induce a preference for sweet taste and lead to obesity later in life. The consumption of whole fruits is recommended for health reasons because they contain—unlike fruit puree—not only fructose and vitamins but also dietary fiber. Fruit juice is a source of free sugars and is therefore considered a potential risk factor for obesity [18,19]. Unfortunately, the consumption of fruit juice is still perceived as healthy by the general public, which is shown by our study, where 30% of all participants stated that they regularly consume fruit juices/smoothies/pouches throughout the ages in order to do something good for themselves. This is in line with an Italian study [11], where 18% of the parents claimed to buy vitamin-enriched fruit juices every time they go shopping, and 38% of the parents buy them often for their children. However, the long-term outcome of fruit pouches has been insufficiently studied [20], but it seems to be clear that children-specific foods are promoted as being healthy and are significantly more expensive, but the nutritional profile of “kids’ food” is less favorable (e.g., higher in fat and carbohydrates but lower in protein) than in the regular version of the product [21].

In our study, 60% of all participants take regular supplements to promote health, vitamins being the second most popular supplement (17% of children under 3 years and 30% of children over the age of 4 years). The prevalence of vitamin and mineral supplement use has increased over the last two decades and is common among adults in North America, where over 70% of adults use supplements [22]. For American children, a prevalence of 33% for regular supplements was found [23], with multivitamins being the most popular one, followed by minerals. A European study revealed that, even in a resource-rich society with free access to any type of food, it may be difficult for some children to reach an adequate vitamin intake, especially for iodine and vitamins D and E [24]. However, the vitamin status of a child depends on many factors. Children do not benefit from random supplementation; in contrast, an inappropriate vitamin intake may cause significant side effects [25]; therefore, supplementations should be guided by a specialist, unlike our participants.

A recent study [26] identified 6.8% of Japanese children as dietary supplement users: amino acids/protein were the most common ones, followed by fish oil, probiotics, and multivitamins. The authors found a significant association between supplement use and a high frequency of sport participation, which was not found in our Swiss cohort. About 25% of children of all ages, who receive special food/supplements to promote health in our cohort, are supplemented with protein. In adolescents and adults, this is a known phenomenon. In the last decade, the idea of beauty shifted for women from a very slim to a more athletic, muscular body, which is reflected in a shift in the gender composition of protein consumers from male dominant to gender-neutral [27]. A Canadian study showed that comparison to idealized bodies on Instagram resulted in lowered confidence and increased body dissatisfaction in young women [28]. Rodgers et al. [29] used a biopsychosocial model to prove in a study with over 700 adolescents that the use of social media was correlated with lower self-esteem, a higher tendency to compare appearance, dietary restraint, and muscle change behaviors among both genders. This is in line with our findings, where a statistically significant correlation between a higher screen time and the use of supplements in all ages was found. Surprisingly, protein (enriched yogurts, drinks, cottage cheese, etc.) is also the second most popular supplement in our cohort of children under the age of 3 years. Literature regarding the benefit of protein fortification in malnourished or premature children is available, also for children from low-income countries [30], but no data are available on the long-term effect of extra protein intake in toddlers. We interpret this new phenomenon in the context of the current dietary trend, which is transferred from parents to children. A different explanation for this correlation might be that children with a higher screen time are more prone to be influenced by the commercials for kids’ food.

Another interesting finding in our cohort is the statistically significant correlation between higher parenteral education and the purchase of supplements/special foods to increase health. It is well known that a lower socioeconomic position is associated with more exposure to unhealthy food, which might lead to obesity and a shorter life expectancy [31]. People with a higher socioeconomic background do not only have access to a more healthy, balanced diet, but the consumer of such a diet is also perceived by other people as generally successful in life [32]. In other words, our diet can be showcased as a status symbol [33,34]—food as classism. On the one hand, the eating behavior of the children clearly reflects what their parents eat [35]; on the other hand, most consumers associate high prices with high quality [36]. Therefore, it seems clear that parents with better financial possibilities spend more money on feeding children with special kids’ packaging and vitamins, believing more expensive food is healthier, although this does not correspond to medical facts, as discussed above. Not only the dietary trend of increased protein intake seems to have caught on with children, but also a shift to the consumption of more plant-based drinks. Although over 90% of families still consume cow’s milk, over 30% of participants also report to regularly consume vegan milk at home. Sousa et al. [37] showed that in Switzerland, the consumption of non-dairy plant-based beverages increased by 19% from 2011 to 2016, while the consumption of cow’s milk decreased by 6%. Schiano et al. [38] showed that a generally negative media messaging toward dairy milk and a positive messaging toward plant-based milks contributes to this trend, even if consumers are not explicitly aware of their own perception change. Negative associations with cow’s milk included hormones, antibiotics, animal welfare, ecological sustainability, intolerances, and allergies. Plant-based drinks are manufactured from a diverse range of nuts, seeds, vegetables, grains, and fruits; therefore, the ingredient profile varies substantially, but most drinks have the tendency to be low in protein, calcium, and other nutrients compared with cow’s milk [39]. Consequently, children whose energy intake largely consists of plant-based drinks should be medically accompanied/advised to avoid nutritional deficiencies and ensure growth and development [40].

Only 4.9% of all filled-out questionnaires were in a language that is not one of the national languages in Switzerland (besides German, French and Italian are the official languages of Switzerland). The aim was to perform a representative survey of what children consume on a regular basis; therefore, a clear, simple questionnaire has been designed to avoid any (cultural) misunderstandings. However, a weakness of this survey is that it is not based on medical data but only on children/parents’ subjective perceptions of how their health can be improved.

## 5. Conclusions

This Swiss cross-sectional study of over 1900 participants reveals that 58% of all participants buy supplements or special kids’ food to promote the child’s health. There is a correlation between higher screen time, higher parental income, and the usage of supplements. A total of 23% of participating families consume at least one plant-based drink on a regular basis. As more and more families use supplements, the pediatrician should not only focus on weight, which reflects the intake of macronutrients but should also take a history of whether children omit certain foods or take supplements to ensure the child does not have a deficiency of micronutrients.

## Figures and Tables

**Figure 1 children-09-01842-f001:**
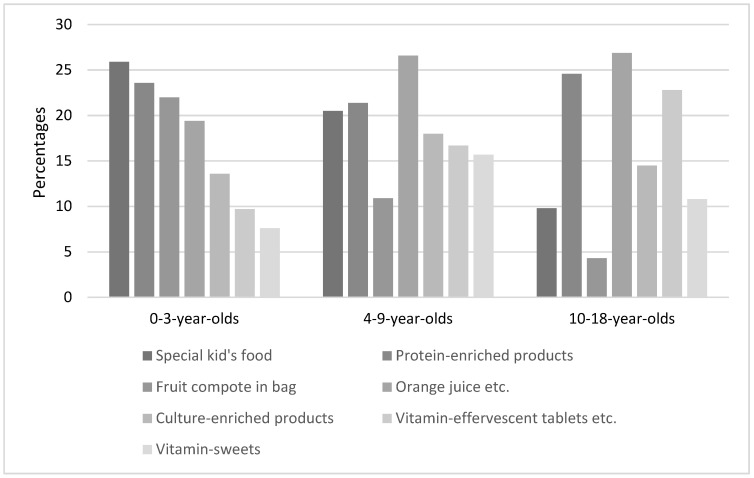
Percentages of supplements/food bought to promote health per age group.

**Figure 2 children-09-01842-f002:**
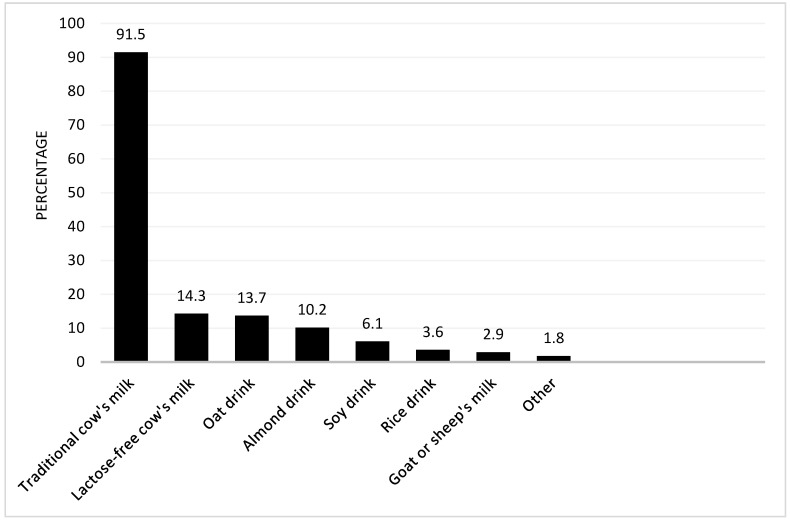
Milk or milk alternatives regularly consumed in the family. Other: coconut milk, raw cow’s milk, buffalo milk, millet drink, pea drink.

**Figure 3 children-09-01842-f003:**
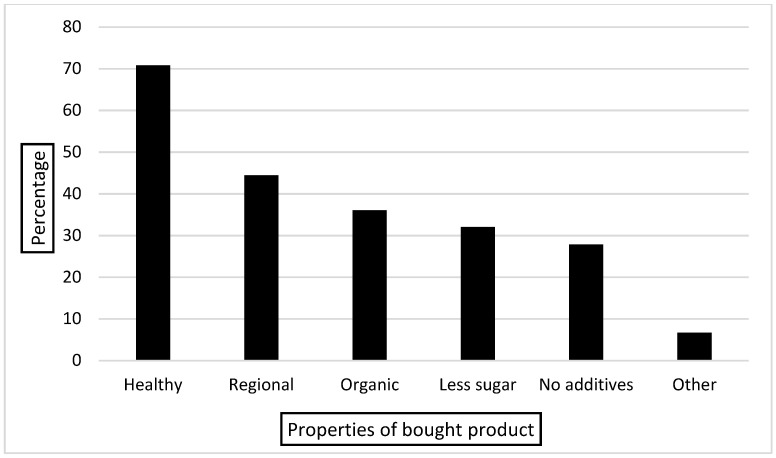
Important properties of bought products. Other: enjoyable (child-oriented), unprocessed (fresh, from own garden/farm, directly from producer, self-made), balanced (diverse, everything in moderation), free or low on certain food or food component (lactose, sugar, peanut, glutamate, palm oil, caffeine, pork, salt, protein, histamine), seasonal, vegetarian/vegan, Demeter, affordable, origin, nutrient content, whole wheat, quality, hygiene, no Nestlé, calorie and fat content.

**Table 1 children-09-01842-t001:** Characteristics of participants (n = 1958).

	N (%)	Buy Supplements/Food to Promote Health (%)
Age group 0–3-year-olds 4–9-year-olds 10–18-year-olds	568 (29) 688 (35.1) 702 (35.9)	315 (55.5) 424 (61.6) 392 (55.8)
Sex Male Female	1057 (54) 901 (46)	594 (56.2) 537 (59.6)
Family background Swiss Abroad Central and Western Europe Eastern and Southeastern Europe Middle East and Central Asia Southern Europe South, East, and Southeast Asia Africa Other (North, Middle, and South America, Northern Europe, and Oceania) Missing answer	864 (44.1) 1039 (53.1) 295 (15.1) 282 (14.4) 107 (5.5) 101 (5.2) 80 (4.1) 77 (3.9) 97 (5) 55 (2.8)	489 (56.6) 614 (59.1) 169 (57.3) 180 (63.8) 63 (58.9) 56 (55.4) 45 (56.3) 39 (50.6) 62 (63.9)
Highest education of parents ≤College, vocational or intermediate diploma school * >College, vocational or intermediate diploma school ** Education status not stated or unknown	763 (39) 1118 (57.1) 77 (3.9)	446 (58.5) 647 (57.9)
Place of residence Urban (>10,000 inhabitants) Rural (<10,000 inhabitants) Missing answer	930 (47.5) 1024 (52.3) 4 (0.2)	548 (58.9) 581 (56.7)
Known allergies Yes No Missing answer	1729 (88.3) 173 (8.8) 56 (2.9)	1009 (58.4) 95 (54.9)

* Less than 7 years of school, compulsory school, pre-apprenticeship, professional apprenticeship, professional school. ** Higher education course, higher professional education, university, technical college.

**Table 2 children-09-01842-t002:** Screen time and physical activity.

**Screen Time and General Activity in 0–3-year-olds (n = 568)**		
	**n (%)**	**Buy Supplements/Food to Promote Health (%)**
Screen time (hours/week) ≤5 >5 Missing answer	432 (76) 118 (20.8) 18 (3.2)	221 (51.2) 82 (69.5)
General activity (hours/week) Yes ≤4 >4 No Missing answer	353 (62.1) 136 (23.9) 191 (33.6) 192 (33.8) 23 (4)	211 (59.8) 83 (61) 110 (57.6) 96 (50)
**Screen time, activity in sports clubs, and general activity in 4–9-year-olds (n = 688)**		
	**n (%)**	**Buy supplements/food to promote health (%)**
Screen time (hours/week) ≤5 5–10 >10 Missing answer	341 (49.6) 195 (28.3) 134 (19.5) 18 (2.6)	201 (58.9) 125 (64.1) 90 (67.2)
Activity in sports club (hours/week) Yes ≤2 >2 No Missing answer	412 (59.9) 257 (37.4) 146 (21.2) 260 (37.8) 16 (2.3)	259 (62.9) 159 (61.9) 93 (63.7) 159 (61.1)
General activity (hours/week) Yes 0–2 2–4 >4 No Missing answer	620 (90.1) 144 (20.9) 212 (30.8) 241 (35) 53 (7.7) 15 (2.2)	394 (63.5) 89 (61.8) 139 (65.6) 151 (62.7) 25 (47.2)
**Screen time, social media account, activity in sports clubs, and general activity in 10–18-year-olds (n = 702)**		
	**n (%)**	**Buy supplements/food to promote health (%)**
Screen time (hours/week) ≤10 10–20 >20 Missing answer	249 (35.5) 282 (40.2) 150 (21.4) 21 (3)	124 (49.8) 167 (63.7) 93 (62)
Social media account Yes No Missing answer	439 (62.5) 243 (34.6) 20 (2.85)	253 (57.6) 132 (54.3)
Activity in sports club (hours/week) Yes 0–2 2–4 >4 No Missing answer	478 (68.1) 136 (19.4) 197 (28.1) 137 (19.5) 206 (29.3) 18 (2.6)	266 (55.6) 66 (48.5) 109 (55.3) 88 (64.2) 119 (57.8)
General activity (hours/week) Yes 0–2 2–4 >4 No Missing answer	594 (84.6) 213 (30.3) 200 (28.5) 163 (23.2) 87 (12.4) 21 (3)	337 (56.7) 125 (58.7) 117 (58.5) 87 (53.4) 47 (54)

**Table 3 children-09-01842-t003:** Multivariable analysis for identifying the potential association between age, education of parents, screen time, general activity of the child, child’s activity in sports club, the use of social media, and the purchase of supplements or food to promote health in the different age groups.

Age Group	Characteristics	OR (95% CI)	*p*
0–3-year-olds	Age	1.24 (1.03–1.49)	<0.05
	Higher education	ns	ns
	Screen time > 5 h	2.45 (1.52–4.05)	<0.001
	General activity	ns	ns
4–9-year-olds	Age	0.9 (0.81–0.99)	<0.05
	Higher education	ns	ns
	Screen time 5–10 h	ns	ns
	Screen time > 10 h	1.56 (1.01–2.44)	<0.05
	Activity in sports club	ns	ns
	General activity	2.44 (1.33–4.54)	<0.01
10–18-year-olds	Age	ns	ns
	Higher education	1.51 (1.09–2.09)	<0.05
	Screen time 10–20 h	1.57 (1.07–2.3)	<0.05
	Screen time > 20 h	1.92 (1.17–3.19)	<0.05
	Activity in sports club	ns	ns
	General activity	ns	ns
	Social media account	ns	ns

## Data Availability

All completed questionnaires are stored in the Children’s Hospital Aarau.

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
