# Peer review of "Use of Health-Promoting Food and Supplements in Swiss Children"

_children, 2022, doi:10.3390/children9121842_

Round 1

Reviewer 1 Report

Section 2: Materials and Methods

- Line 56: No information is presented on if and how the questionnaire was validated. If the questionnaire is not validated, the results must be presented with caution as they can only be observational and this should be stated in the paper.

- Line 58: Given that the Questionnaire has been translated in many non-European languages, the population is clearly multi-cultural. It should be described how the cultural differences have been addressed in the treatment of the results to avoid any bias or justified that there was no bias.

- Line 72: Given the importance of the results it should be defined and described what kind of products the category 7, protein-enriched products, contains and how respondents were informed about these products. 

Section 3: Results

- Table 1 - Section 4 (Highest education parents): Data lines are not aligned.

- Line 110: typo: ‘or was’

- Figure 1: It makes more sense to present the lowest age group as first group, followed by the older age groups.

- Table 2: It is not explained what is the basis for deciding the cut off value for screen time (for the 0-3 years group it is 5h, for the 9-8 years group 10h and for the 10-18 years group 10-20h).

- Line 125: Best indicate that general activity is that of the child and not that of the parents.

- Figure 2: The visual is that of figure 3. Both have been switched.

- Figure 3: The visual is that of figure 2. Both have been switched.

- Figure 3: It is not explained how the different properties have been identified by the respondents.

Section 4: Discussion

- Line 150: What does ‘regularly’ mean? Once a day/week?

- Line 153: Special kid’s food are defined as fruit pouches, biscuits and crackers. Where do milk-based drinks specifically formulated for 2-3 year olds fit? Have they been left out?

- It is noted that the literature cited is not based on a systematic literature search. No information is provided on in- and exclusion criteria of relevant papers.

- Nearly all of the references cited investigated children’s food intake and behaviour in other regions of the world. These studies are carried out in Australia (ref 9,18), New Zealand (ref 12), Japan (ref 23), and US (ref 10,11,13,16, 19, 20. 28). Aspects such as cultural background, food consumption and composition, social media and physical activity habits and patterns, etc may be fundamentally different from those in Switzerland. Still, these studies are used to draw conclusions in the context of Switzerland. For the discussion of the results to be relevant, studies carried out in a European setting should be referenced.

- Based on the reference, it is doubtful that conclusions can be drawn such as:

o Besides beliefs and values, mainly costs, availability and parent’s capacities influence the choice (line 158) – Austrailan data

o Health might be compromised for convenience (line 159) – US data

o This is the secret of its success: Children like the colorful sweets and parents are happy to purchase it because it is convenient and the product suggests to them that it is 167 healthy because it contains fruit (Line 166) – US data

o Median total sugar content of kid’s pouches was more than three times that of cans and jars – US data

o Children-specific foods are promoted as being healthy, are significantly more expensive but the nutritional profile of "kid`s food" is less favorable (e.g. higher in fat, carbohydrate but lower in protein) than in the regular version of the product – Australian data

o Etc

- Line 220: Supplements wth protein. No information is provided on the nature of such products. Given the results it seems important to describe what products are meant and how they have been defined by or for the respondents.

- Line 212: protein: No information is provided on the nature of such products. Given the results it seems important to describe what products are meant and how they have been defined by or for the respondents.

- Line 223: Unclear sentence to be revised.

- Line 231: Shift to a more vegan diet: It is not clear how that conclusion can be drawn solely on the fact that consumers consume plant-based milks, even if 30% consume such products. Are such consumers only using plant-based drinks or in combination with milk in their diet?

Section 5: Conclusions (lines 244-250)

- It seems that the conclusions drawn are not based on the study results and should therefore be revised:

o Diet is not only subject to trends that reflect the current zeitgeist and is transferred to the children, it can also be a status symbol. – This is hard to conclude from the results as zeitgeist, status symbol and transfer to children were not investigated.

o A higher socioeconomic status does not equal a healthier, more balanced diet, but rather the freedom to purchase more expensive products with supposedly better-quality content for oneself and one’s child. – This is hard to conclude from the results as product composition, healthiness and prices of the foods are not reported.

o As more and more families use supplements, the pediatrician should not only focus on weight, which reflects the intake of macronutrients, but should also take a history of supplements, which, despite good intentions, may not only do good when taken. – This may be the opinion of the authors and it is probably true, but this cannot directly be concluded from the results of the study.

- Also the text of the abstract should be revised accordingly.

Reviewer 2 Report

The authors presented the findings of an investigation into the eating patterns of Swiss children aged 0 to 18 years. Although the manuscript is written and presented in a high quality, the content and the scientific message could be improved to be regarded as a research article.

The authors evaluated dietary habits in Swiss children aged 0 to 18 years. They asked participants if they bought supplements/foods to promote the health of the child / adolescents, about the consumption of milk / milk alternatives in the family, and what properties motivated them to buy this type of products. Using data such as age, sex, place of birth of the child and the parents, population of residence, the level of education of both parents, the amount of screen time, the use of a social media account, and general physical activity and activity in a sport club, underlying disease, and known allergies, the researchers tried to find out if there is any correlation between these data and dietary habits.

The results of the research mostly consisted in descriptive analysis of the participants (age group, sex, family background, highest education of parents, place of residence, allergies, screen time, general activity, etc.).

In all age groups, the authors identified correlations between the purchase of supplements or food to promote health and screen time. Is it about parents' or children's screen time? Could this be linked to the response to food marketing?

In the groups of children under 3 years and 4-9 years old, the authors identified correlations between the purchase of supplements or food to promote health and the age of the child. At this age, for most children, we can talk about the eating habits of parents, but unfortunately, we do not have a statistical correlation between parents’ education (or family income) and their children’s eating habits.

In my opinion, it does not add anything to the subject area compared to other published material.

The authors should consider conducting a more detailed analysis of the characteristics and eating habits of parents who have children who cannot make their own decisions regarding their eating habits (e.g., small age). In addition, family income could be a factor that influences eating habits.

The conclusions are general and are not consistent with the evidence and arguments presented.

The references are appropriate.

Tables and figures are representative of the description of the characteristics analyzed.

Round 2

Reviewer 1 Report

The manuscript has significantly improved.

From the second version submitted, it was not possible to see if and how the figures and tables had been updated. This requires checking on the final proof.

Author Response

Dear Reviewer 1, 

Unfortunately I could not paste a tracked-version of the tables/figures into the required template. All tracked versions of tables/figures can be send by email on request.

Warm regards, 

Corinne Légeret

Reviewer 2 Report

The revised version of the manuscript, which includes the addition of new information and further explanations, clarifies the ambiguous aspects presented in the initial manuscript and improves its scientific quality.

However, it is necessary to clarify and detail the correlation between higher screen time of the children between 0 and 3 years of age, and the purchase of supplements/food to promote health in this group. Is there scientific evidence that a 6-month-old baby can request/choose a certain food after seeing it on television? Perhaps the lower age limit for this group of children should be adjusted to give more credibility to the research results.

Author Response

Dear Reviewer 2, 

Thank you for this hint. It is an important point that has now been added to the discussion section.

Warm regards, 

Corinne Légeret